# Factors associated with caring behaviors of family caregivers for patients receiving home mechanical ventilation with tracheostomy: A cross-sectional study

**Hyang Sook Kim[1,2], Chung Eun Lee[1,2], Yong Sook Yang ![ORCID][1]***

**1** College of Nursing Brain Korea 21 FOUR Project, Yonsei University, Seoul, Republic of Korea,
**2** Severance Hospital, Seoul, Republic of Korea

\* ysyang100061@gmail.com

**Data Availability Statement:** All relevant data are within the manuscript and its Supporting information files.

## Abstract

### Background

The number of patients on home mechanical ventilation (HMV) worldwide has been steadily rising as medical technological advanced. To ensure the safety and quality care of the patients receiving HMV with tracheostomy, caring behavior of family caregivers is critical. However, studies on caring behavior of family caregivers and its associated factors were remained unexplored. This study aimed to describe the caring behaviors of family caregivers for patients receiving home mechanical ventilation with tracheostomy and to identify factors associated with their caring behaviors.

### Methods

This was a cross-sectional study for 95 family caregivers for patients with invasive home mechanical ventilation in South Korea. Caring behaviors were assessed by the Caring Behavior Scale with 74 items with 5-point Likert scale. Data were analyzed using multiple regression analysis.

### Results

Caring behaviors score of caregivers was 304.68±31.05 out of 370. They were significantly associated with knowledge on emergency care (β = 0.22, $p$ = .011), number of required instruments for care (β = 0.21, $p$ = .010), frequency of home visit care (β = 0.19, $p$ = .017), experience of emergency situation for the last six months (β = 0.19, $p$ = .009) and activities of daily living of patient (β = 0.27, $p$ = .002).

### Conclusion

Development of standardized multidisciplinary discharge education for improving the caring capacity of caregivers is required for successful and healthy application of home mechanical ventilation.

**Funding:** This research received no specific grant from any funding agency in the public, commercial, or not-for-profit sectors.

**Competing interests:** The authors have declared that no competing interests exist.

## Introduction

The number of patients on home mechanical ventilation (HMV) worldwide has been steadily rising as medical technological advanced [1]. HMV is a medical treatment for long-term breathing support for patient who cannot keep their airway patent by themselves but need to be discharged from the emergency setting, such as patients with neuromuscular diseases or respiratory problems [1]. HMV is administered as either invasive positive pressure ventilation (IPPV), which involves a tracheostomy, or noninvasive positive pressure ventilation (NIPPV), which uses a mask or nasal intubation [2]. HMV with tracheostomy is generally administered 24 hours a day (i.e., much longer than for NIPPV) [3]. As a result of the characteristics of the treatment and their illnesses, patients receiving HMV with tracheostomy have limited physical activity, which makes them highly dependent on caregivers [4].

The global prevalence of HMV was reported to be 6.6–23 per 100,000 people [1]. In South Korea, HMV prevalence has also gradually increased, reaching 9.3 per 100,000 people in 2017 [5]. Recently, 83.3% of patients decided to be discharged with home mechanical ventilation after finishing acute care in South Korea [6,7]. It is mainly for spending time at home to maintain quality of life for both family and patients [8,9] and to lessen the burden of healthcare costs [10]. It is also partly related to the South Korea's National Health Insurance Service, which was broadened its scope to the use of HMV to cover pulmonary diseases and cardiac diseases with chronic respiratory symptoms in 2016 [11]. However, the readmission rate of patients with HMV was as high as 60%, with many of these cases returned to ICUs due to the worsened or emergency situations [12]. Patients with long-term HMV with tracheostomy showed higher readmission rates than NIPPV patients [13] and the cause of their readmission was associated with safety issues with HMV in 76% of cases [13]. Consequently, to ensure the safety of the patients receiving HMV with tracheostomy, family caregivers' awareness of emergency situations, coping skills, and ability to manage equipment play key roles [14].

To improve caregivers' competencies for dealing with safety issues of patients on HMV with tracheostomy, specialized education programs are needed as home-based caregivers are required not only to meet patients' physical needs but also to perform nursing tasks such as inner cannula change, airway secretion suctioning, and positive pressure ventilation using an Ambu-bag [15]. Current pre-discharge caregiver education in Korea is provided without a prior plan, mostly by ventilator rental company employees using leaflets [16]. Moreover, caregiver education is focused mainly on the skills required for home ventilator management, not on communication with patients who are unable to speak and rehabilitation of those who are bedridden [17]. Thus, the education caregivers currently receive is insufficient for them to acquire the necessary skills to independently deliver safe care for patients [18].

Many previous studies conducted identified knowledge on emergency care of caregivers of patients receiving HMV with tracheostomy [19], frequency of caregivers' caring behaviors [20], family caregivers' caring experience for patients with HMV, the burden of caring behaviors, and quality of life of caregivers [21,22], and the correlation between airway suctioning and the occurrence of pneumonia [23]. Studies on associated factors of caring behaviors of caregivers were remained unexplored. Therefore, this study aimed to identify the factors that affect the caring behaviors of family caregivers of patients receiving HMV with tracheostomy, and to suggest interventions for improving their care performance.

## Materials and methods

### Study design and participants

This is a descriptive, cross-sectional study to explore the caring behaviors of family caregivers for patients on HMV with tracheostomy, and to determine the factors associated with these caring behaviors. The participants in this study were family caregivers who provided home-based care for at least six hours a day to patients receiving HMV with tracheostomy. Participants were approached by home visit nurses using the client list of the ventilator rental company based in Seoul, Korea. The inclusion criteria were: 1) aged equal to or greater than 19 years old and having a family relationship with the patient, such as a spouse, parent, or child (including daughter-in-law and son-in-law); and 2) living with the patient and providing primary, unpaid care. Caregivers of patients with NIPPV or impaired cognitive and physical functions were excluded.

### Measures

**Characteristics of the participants.** The questionnaires on participants' general characteristics was developed based on a previous study that investigated the association between the features of care provided by family caregivers and the burden they experience [21]. The general characteristics of interest included gender, age, and education level were assessed. Care related variables included discharge education experience, experience of emergency situations for the last six months, satisfaction with home visit care, preparedness of required equipment for emergency situations, and difficulties in caring patients. Additionally, an open-ended question was added to describe participants' experiences with ventilator-related emergency situations (S1 File).

Knowledge of emergency care of caregivers was assessed using the Korean version of the Knowledge of Emergency Care Scale developed by Hwang et al. [17] (S1 File). It was originally developed by Kun et al. [19] to measure the capability of emergency care and response which should be possessed by primary caregivers of patients receiving HMV with tracheostomy. Permission to use this tool was obtained from the original author and also from the author who developed the Korean version. It consists of five categories with 25 items: aspiration management (4 items), routine management of tracheostomy tubes (6 items), ventilator management (5 items), management when the tracheostomy tube is decannulated (5 items), and basic cardiopulmonary resuscitation (5 items). One point is awarded for correct answers and zero point for incorrect answers; thus, the total score range is 0–25, with higher scores indicating higher knowledge level. Cronbach's α of the original questionnaire was.91 [17], while it was.95 in this study.

**Characteristics of the patients.** Patient information included general characteristics, disease manifestations such as diagnosed disease, level of physical dependency, HMV-related information (i.e., duration, ventilator mode, use of oxygen, application hour per day), frequency of home visit care, and activities of daily living (ADLs). Their ADLs were evaluated using a 12-item scale, which indicates patient dependency on caregivers (S2 File). Each item of ADLs was scored using a three-point Likert scale (1 = "*can do it alone*," 3 = "*totally depends on help*"); thus, total scores ranged from 12 to 36 points. Higher summed score means lower level of ADLs which indicates higher dependency on their caregivers [24]. Cronbach's α in this study was.95.

**Caring behaviors.** Caring behaviors of the caregivers were measured using the Caring Behavior Scale developed by Hwang et al. [21] after gaining a permit to use. It comprises 10 categories with 74 items: HMV management (14 items), airway management (11 items),

nutrition management (10 items), rehabilitation exercise (3 items), personal hygiene (6 items), commodity management (4 items), environmental management (4 items), communication (2 items), psychological nursing (2 items), and aspiration management (18 items). Items were scored by a five-point Likert scale (1 = "*not at all*," 5 = "*always*") (S3 File). The higher summed score means the higher ability to provide care. Cronbach's α of the original questionnaire was.91 [17], while it was.90 in this study.

## Ethics statement

This study was approved by the institutional review board of the Yonsei University Health system, Severance Hospital (# 4-2019-0230) where the researchers were working. Home visit nurses affiliated with the ventilator rental company conducted in-person data collection from family caregivers of patients receiving HMV with tracheostomy. During the home visits, nurses provided detailed information of the study and caregivers freely decided their participation. Caregivers were informed that their participation or refusal would never related to home visit care. The research was conducted only with caregivers who provided written informed consent. A small gift was given to all participants when they finished the questionnaires.

## Data collection process

Data were collected from May to July 2019 using self-report questionnaires. Five home visit nurses working for the ventilator rental company collected the data during their home visits. The home visit nurses had 2-hour training for data collection prior to starting the survey. The home visit nurses explained the purpose, methods, and procedures of the study to the caregivers and obtained informed consents. The nurses also provided supports for family caregivers who were illiterate or had problems with seeing vision by reading the questions and recording the responses. Based on the sample size estimated for multiple linear regression analysis with a 10% dropout rate, 130 caregivers were initially approached, with questionnaires distributed to the 100 who agreed to participate. Of these 100, after excluding the questionnaires of five caregivers whose responses were insufficient, the data of 95 caregivers were analyzed.

## Statistical analysis

The data (S4 File) were analyzed using the SAS 9.4 program. Frequencies and percentages, and means and standard deviations were used for descriptive analysis. Differences between groups were compared using independent *t*-tests and analyses of variance followed by the post-hoc multiple comparison. Correlations between continuous variables and care behaviors were analyzed using Pearson's correlation coefficients. The factors associated with care behaviors of the family caregivers were identified by multiple regression analysis, and the variables for final regression model were selected by the backward selection method.

## Results

### Characteristics of family caregivers

A total of 95 family caregivers participated. The mean age of the family caregivers was 51.02 (± 10.85) years. Out of 95 participants, 71 (74.7%) were female, and 78 (82.1%) were married. In terms of education level, 56 (59.0%) participants were graduated from their college. Regarding the relationship with their patient, 54 (56.8%) caregivers were parents, and 28 (29.5%) were spouses of patients. Seventy-eight (82.1%) participants were unemployed, and 35 (36.8%) reported that the monthly household income was over 3 million Korean Won (1,000 Korean Won = $1), and 19 (20.0%) reported that they earned less than 1 million Korean Won per

month. Regarding care-related characteristics, 85 (89.5%) reported that they had care-related education in the hospital at the time of the patient's discharge. The mean number of home visit nursing care per month was 1.12 (± 0.86). Eighty-seven (91.6%) responded that they had no experience of emergency situations for the last six months. The mean knowledge level on emergency care was 20.64 (± 2.14) out of 25, and the mean score for Satisfaction with Medical Service was 5.24 (± 2.68) out of 10 (Table 1).

## Characteristics of patients

The mean age of the patients was 38.08 (±26.2) years old, this relatively low mean age was owing to the inclusion of 30 children in the sample. Of the 95 patients, 65(68.4%) were male. Regarding underlying diseases, 67 (70.5%) patients had neuromuscular diseases while 15 (15.8%) had respiratory diseases. Among 95 patients, 33 (34.7%) had been on HMV for 3–4 years, 28 (29.5%) for less than three years, 19 (20.0%) for 5–10 years, and 15 (15.8%) for more than 10 years. Regarding the modes of HMV, 70 (73.7%) were on pressure-controlled ventilation (PCV) and 25 (26.3%) on volume-controlled ventilation (VCV). Among 95 patients; 34 (35.8%) used oxygen. The mean hour of daily HMV usage was 21.34 (± 6.16), and patients' physical dependency (i.e., the ADL scale) was 22.47 (± 4.02). The average required equipment was 12.73±1.4; this included an Ambu bag, oxygen monitor, blood pressure monitor, thermometer, spare cannula and circuit, cough inducer, extra battery, and extra portable suction machine, which were needed in case of emergency at home. Characteristics of the patients are presented in Table 2.

**Table 1. General characteristics of family caregivers (*n* = 95).**

| Characteristics | Categories | *n* (%) | Mean±SD |
|---|---|---|---|
| Age (years) | | | 51.02±10.85 |
| Sex | Female | 71 (74.7) | |
| | Male | 24 (25.3) | |
| Marital status | Married | 78 (82.1) | |
| | Unmarried | 12 (12.6) | |
| | Others | 5 (5.3) | |
| Educational level | ≤High school | 39 (41.1) | |
| | ≥College | 56 (59.0) | |
| Employment status | Yes | 17 (17.9) | |
| | No | 78 (82.1) | |
| Monthly income (Korean Won) | <1 million | 19 (20.0) | |
| | 1–<2 million | 21 (22.1) | |
| | 2–<3 million | 20 (21.1) | |
| | ≥ 3 million | 35 (36.8) | |
| Relationship with patients | Spouse | 28 (29.5) | |
| | Parents | 54 (56.8) | |
| | Others | 13 (13.7) | |
| Education for discharge care | Yes | 85 (89.5) | |
| | No | 10 (10.5) | |
| Experience of emergency situation (within 6 months) | Yes | 8 (8.4) | |
| | No | 87 (91.6) | |
| Knowledge on emergency care(0–25) | | | 20.64±2.14 |

**Table 2. General and disease-specific characteristics of patients (n = 95).**

| Characteristics | Categories | n (%) | Mean±SD |
|---|---|---|---|
| Age (years) | | | 38.08±26.62 |
| Sex | Male | 65 (68.4) | |
| | Female | 30 (31.6) | |
| Primary disease | Neuromuscular | 67 (70.5) | |
| | Lung/Airway | 15 (15.8) | |
| | Others | 13 (13.7) | |
| Duration of HMV[a] | <3years | 28 (29.5) | |
| | 3–4years | 33 (34.7) | |
| | 5–9years | 19 (20.0) | |
| | ≥10years | 15 (15.8) | |
| Ventilator mode | Volume control | 25 (26.3) | |
| | Pressure control | 70 (73.7) | |
| $O_2$ usage | Yes | 34 (35.8) | |
| | No | 61 (64.2) | |
| HMV[a] application hours/day | | | 21.34± 6.16 |
| Number of home visits/month | | | 1.12± 0.86 |
| [b]ADL | | | 22.47±4.02 |
| Number of required equipment | | | 12.73±1.4 |

[a]HMV = home mechanical ventilator,

[b]ADL = activities of daily living.

## Caring behaviors of family caregivers

The overall summed score of the caring behaviors was 304.68 (± 31.05) out of 370. Caring behavior of ventilator management was 61.94 (±6.94) out of 70 while airway management was 48.04 (±5.70) out of 55. Nutritional support was scored at 37.93 (±17.03) out of 50, rehabilitation 7.84 (±3.34) out of 15, and the personal hygiene 26.54 (±3.56) out of 30. Equipment management was scored at 18.55 (±2.26) whereas environmental management was 19.01 (±1.96) out of 20. Communication with patients was scored at 6.43 (±3.02) out of 10. Suctioning showed 78.36 (±7.14) out of 90 (Table 3).

There was no significant difference of care behaviors according to the general characteristics of family caregivers (Table 4).

**Table 3. Caring behavior scores of family caregivers.**

| Categories | Mean±SD | Min | Max |
|---|---|---|---|
| Total caring behavior | 304.68±31.05 | 72 | 370 |
| Ventilator management | 61.94±6.94 | 14 | 70 |
| Airway management | 48.04±5.70 | 11 | 55 |
| Nutrition support | 37.93±17.03 | 10 | 50 |
| Rehabilitation | 7.84±3.34 | 3 | 15 |
| Personal hygiene | 26.54±3.56 | 6 | 30 |
| Equipment management | 18.55±2.26 | 4 | 20 |
| Environmental management | 19.01±1.96 | 4 | 20 |
| Communication | 6.43±3.02 | 2 | 10 |
| Suction | 78.36±7.14 | 18 | 90 |

**Table 4. Difference of caring behavior scores by characteristics of family caregivers.**

| Characteristics | Categories | Caring behaviour | | |
|---|---|---|---|---|
| | | Mean±SD | t or F | p-value |
| Sex | Male | 298.29±31.54 | 1.37 | .245 |
| | Female | 306.85±30.80 | | |
| Marital status | Married | 311.83±24.00 | 1.55 | .218 |
| | Unmarried | 302.33±32.46 | | |
| | Others | 324.20±6.69 | | |
| Educational level | ≤High school | 309.10±33.84 | 1.34 | .249 |
| | ≥College | 301.61±28.85 | | |
| Employment status | Yes | 308.59±29.99 | 0.33 | .570 |
| | No | 303.83±31.40 | | |
| Monthly income (Korean Won) | <1 million | 296.68±33.42 | 1.26 | .293 |
| | 1–<2 million | 315.14±24.52 | | |
| | 2–<3 million | 302.80±35.12 | | |
| | ≥3 million | 303.83±30.40 | | |
| Relationship | Spouse | 305.36±34.57 | 1.15 | .320 |
| | Parents | 301.59±31.45 | | |
| | Others | 316.08±17.52 | | |
| Education for discharge care | Yes | 304.87±31.36 | 0.03 | .866 |
| | No | 303.10±26.78 | | |
| Experience of emergency (within 6 months) | Yes | 322.63±18.39 | 2.98 | .088 |
| | No | 303.03±31.52 | | |

## Factors associated with caring behaviors of family caregivers

To determine the factors predicting caring behaviors of caregivers, independent variables were selected based on the previous studies. In the final model of analysis, caregiver characteristics (i.e., caregiver variables of knowledge on emergency care, frequency of home nurse visits, preparedness to use required equipment during emergency situations, and experience of emergency situations for the last six months) and patients' disease-specific characteristics (i.e., ADL, primary disease, and the number of required equipment) were included. Age and gender of caregivers were adjusted in the model.

The final multiple regression analysis showed that the explanatory power was 48.4%. The results showed that higher patient ADL, higher caregiver knowledge on emergency care, higher preparedness to use required equipment during the emergency situations, greater number of home visits, and having experience of emergency situations within the past six months were significantly associated with the higher caring behaviors of family caregivers (Table 5).

## Discussion

This study aimed to estimate the caring-behavior levels of family caregivers of patients receiving HMV with tracheostomy after hospital discharge, and also sought to obtain basic data that could inform the development of nursing education programs for reducing home caregivers' burden. The results showed that the caring behavior of caregivers were associated with ADL of patient, knowledge on emergency care, preparedness of equipment for emergency situations, and experience of emergency situations.

The study results presented that the family caregivers' characteristics were not significantly associated with the caring behaviors of caregivers, which caregivers provided on a regular

**Table 5. Factors associated with caring behaviors of family caregivers.**

| Variables | B | SE | β | t | *p*-value |
|---|---|---|---|---|---|
| Age | 2.05 | 0.63 | 0.27 | 3.24 | .002 |
| Knowledge on emergency care | 3.15 | 1.21 | 0.22 | 2.59 | .011 |
| Number of home visits per month | 6.77 | 2.77 | 0.19 | 2.44 | .017 |
| Emergency experience (ref: No) | | | | | |
| Yes | 20.58 | 8.67 | 0.19 | 2.37 | .009 |
| ADL[a] of patients | 2.05 | 0.63 | 0.27 | 3.24 | .002 |
| Number of required equipment | 4.63 | 1.77 | 0.21 | 2.62 | .010 |

[a]ADL: Activities of daily living.

basis. Notably, however, the provision of respiratory rehabilitation care was not sufficient, although a previous study reported that respiratory rehabilitation at home is effective for increasing exercise capacity and quality of life [25]. In other research on respiratory rehabilitation in the home, the use of medical equipment such as cough assists, postural drainage, and patient monitoring with pulse oximetry accounted for only 18%, 13.6%, and 16%, respectively, of the care provided [26]. In this study, preparedness of required equipment was associated with a higher level of caring behaviors; thus, assessment of equipment preparedness on discharge and connection of home-based respiratory rehabilitation with home visiting care are needed to improve care at home and help reduce patients' length of hospitalization.

The scores for the family caregivers' knowledge on emergency care was lower than those reported in one previous study [19], but higher than those reported in studies of patients with ALS [17]. This difference may be due to the duration that care was provided, resulting in different levels of knowledge and education over time. However, given the duration of time that our participants had been provided care, we considered that the proportion who had sufficient knowledge to provide appropriately safe care was low [19]. The element of knowledge of emergency care for which the participants scored highest was ventilator management, while the lowest-scoring element was airway suction. The participants' score for airway clearance technique was similar to that reported in previous surveys [19]. In contrast, we found that our participants' ventilator management scores were high. A possible reason for this is that education on ventilator equipment is provided on a regular basis by the rental agent, and an agent is available for consultation via phone at any time of the day or night. Thus, the contents of the education that is provided by rental agents and of the education provided at hospitals at patient discharge should be examined and integrated into operational systems for patient management.

When caring for patients receiving HMV with tracheostomy, airway cleaning techniques and the ability to recognize and manage emergency situations are particularly important [19]. However, previous studies have reported that only 48% of caregivers feel prepared at the time of discharge [27], and a Korea-based study reported that 31% of caregivers do not receive education at patients' initial discharge [28]. In this survey, over 89.5% of the participants received a ventilator care education program offered by the rental agent. One study showed that education regarding providing care for patients receiving HMV with tracheostomy commences immediately after they undergo tracheostomy, and is performed by a multidisciplinary team comprising physicians, nurse practitioners, rapid response teams, social workers, and home health nurses [27]. In particular, the provision of education on emergency situations needs to be based on realistic clinical scenarios, while discharge education must be rooted in the practical needs of family caregivers, including how to properly communicate with patients and

support their long-term rehabilitation. In contrast, in South Korea the main teaching methods include oral presentation and printed handouts, and limited demonstrations of how to provide care for patients using ventilators are performed during rental agent visits; thus, caregivers in South Korea do not receive video-based education. This indicates that a more organized discharge process is needed, including a standardized discharge program and the utilization of various teaching methods at the hospital. In addition, policies for providing safe and efficient home health services should be established for family caregivers of patients receiving HMV with tracheostomy, in terms of supporting hospitalization, mobility, and payment schemes.

This study has several limitations. First, it only considered family caregivers of patients receiving HMV with tracheostomy who were living in the capital city of Seoul and a metropolitan city of Gyeonggi Province, which may affect the generalizability of the findings. Second, this study focused only on determining the extent to which knowledge of emergency situations and general characteristics affect family caregivers' caring behaviors and the level of care they provide. Future research should consider numerous additional variables, such as burdens influencing care level and quality of life, and also include qualitative evaluations of family caregivers' actual caring skills by nurses. Third, the number of patients who had experienced an emergency situation was relatively small, although it turned out to be a statistically meaningful factor in a previous study [29] and this study. Future studies should evaluate the effect of emergency situation experience in larger patient groups. Lastly, there is a possibility of discrepancies between self-reported and actual caring behaviors; thus, there is a need for studies exploring the correlation between family caregivers' self-reports of their care performance and their actual performance.

## Conclusions

Development of standardized multidisciplinary discharge education for improving the caring capacity of caregivers is required for successful and healthy application of home mechanical ventilation. Realistic and specific clinical scenario-based education should be provided to family caregivers right from the time of patients' hospital admission. In addition, it is necessary to create comprehensive and consistently available home health services that are affiliated with home-visit nurses and community resources and are available to caregivers after discharge.

## Supporting information

**S1 File. Emergency knowledge questionnaire.**
(PDF)

**S2 File. ADL of patient questionnaire.**
(PDF)

**S3 File. Caring behaviour questionnaire.**
(PDF)

**S4 File. Data set.**
(PDF)

## Acknowledgments

We would like to thank the home visit nurses for supporting the survey and to Editage (www.editage.co.kr) for English language editing.

## Author Contributions

**Conceptualization:** Hyang Sook Kim, Chung Eun Lee, Yong Sook Yang.

**Data curation:** Hyang Sook Kim.

**Formal analysis:** Hyang Sook Kim, Chung Eun Lee, Yong Sook Yang.

**Investigation:** Hyang Sook Kim, Yong Sook Yang.

**Methodology:** Hyang Sook Kim, Chung Eun Lee, Yong Sook Yang.

**Project administration:** Hyang Sook Kim.

**Validation:** Chung Eun Lee, Yong Sook Yang.

**Writing – original draft:** Hyang Sook Kim, Yong Sook Yang.

**Writing – review & editing:** Hyang Sook Kim, Chung Eun Lee, Yong Sook Yang.

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
