## [Decision Letter · Decision Letter 0]

13 Apr 2021

PONE-D-21-02293

Factors associated with caring behaviors of family caregivers for patients receiving home mechanical ventilation with tracheostomy: A cross-sectional study

PLOS ONE

Dear Dr. YANG,

Thank you for submitting your manuscript to PLOS ONE. After careful consideration, we feel that it has merit but does not fully meet PLOS ONE’s publication criteria as it currently stands. Therefore, we invite you to submit a revised version of the manuscript that addresses the points raised during the review process.

We look forward to receiving your revised manuscript.

Kind regards,

Tai-Heng Chen, M.D.

Academic Editor

PLOS ONE

'..Yong Sook Yang received a scholarship from Brain Korea 21 FOUR Project of Yonsei University College of Nursing.'

'This research received no specific grant from any funding agency in the public, commercial, or not-for-profit sectors.'

Reviewers' comments:

Reviewer's Responses to Questions

**Comments to the Author**

1. Is the manuscript technically sound, and do the data support the conclusions?

Reviewer #1: No

Reviewer #3: Yes

Reviewer #4: Partly

2. Has the statistical analysis been performed appropriately and rigorously? 

Reviewer #1: No

Reviewer #3: Yes

Reviewer #4: Yes

3. Have the authors made all data underlying the findings in their manuscript fully available?

Reviewer #1: Yes

Reviewer #3: Yes

Reviewer #4: Yes

4. Is the manuscript presented in an intelligible fashion and written in standard English?

Reviewer #1: No

Reviewer #3: Yes

Reviewer #4: Yes

5. Review Comments to the Author

Reviewer #1: Thank you for the opportunity to review this descriptive analysis investigating the factors associated with caring behaviors of family caregivers for patients receiving home mechanical ventilation with tracheostomy.

After reading this article carefully, some significant issues should be addressed.

1.This study has used several questionnaires or scales as the measurement tool, for example, the questionnaires for general characteristics of participants including care related variables (line 96-100), Knowledge of Emergency Scale (line 101-102), the ADLs (line 118-119) and the Caring Behavior Scale (line 123-128). However, I cannot find the questionnaires themselves and if the questionnaires have been validated after reading some of the references. Although the questionnaires might be published in other journals, the questionnaires might not be suitable for the participants in this study, and the reliability and validity should be mentioned in this article. For example, I am wondering how does the questionnaire ask about the “education for discharge care”. Does the answer “YES” “NO” really mean the true results that the researcher wants to measure? In addition, the rating method of ADLs seems not the same as the reference [22] mentioned.

2.How did the 100 participants been selected? Selection issue should be considered.

3.The results in line 216 to line 22 should be re-written without directly mention the beta and p-value which the reader can see in Table 5.

Reviewer #3: It is an interesting study for caring behaviors of HMV patients. The study is important for surveying caring quality for caring behaviors. There are several minors questions needed to be answered.

Question 1: 91.6% of patients did not have experience of emergency situation in the study. However, the table 5 showed the factor of experience of emergency situation positively associated with caring behavior. How could author explain the result because most of patients seem to be relative stable?

Question 2: The mean age of patients is young (38.08±26.62) for the need of HMV. The authors should state what kinds of neuromuscular and airway disease of patients.

Question 3: In line 194: “Caring behavior of ventilator management was 61.94 (±6.94) out of 50”. It seems 61.94 (±6.94) out of 70 instead of 50.

Reviewer #4: This study aimed to delineate the caring behaviors of family caregivers for patients receiving home mechanical ventilation(HMV) with tracheostomy and to identify factors associated with their caring behaviors in South Korea. This was a cross-sectional study for 95 family caregivers for patients with invasive HMV. The factors associated with care behaviors of the family caregivers were identified by multiple regression analysis, and the variables for final regression model were selected by the backward selection method. The results showed that the caring behavior of caregivers were associated with ADL of patient, knowledge on emergency care, preparedness of equipment for emergency situations, and experience of emergency situations. This indicates that a more organized discharge process is needed, including a standardized discharge program and the utilization of various teaching methods at the hospital. The author concluded that development of standardized multidisciplinary discharge education for improving the caring capacity of caregivers is required for successful and healthy application of home mechanical ventilation.

Major points:

1.The conclusion part in the page 17 is too subjective because there was no result in this study support using the simulation training with specific scenarios. I would suggest only the first sentence to be kept. Please make point-to-point reply to the results by Tables.

2.There is one measure needed to be clarified. In Table 3 (Page 11). Caring Behavior Scores of Family Caregivers regarding Ventilator management: 61.94±6.94 ( min 14 and Max 50). Why?

3.For readers, it would be interesting to understand the current caregiver training programs before discharge. Could the authors explain more about this? That would help why the score regarding rehabilitation and communication are relatively lower than other domains. And from the results, what kind of education or training program needs to be modified? Please explore more on this for implication.

4. Could the authors disclose the 12.7 items of required equipments in Table 2?

minor points:

1.This is a self-reported questionnaire to reflect the caring behavior score. I wonder if the actual caring skills of family caregivers can be measured by home visited nurses or health care workers at home.

2.This could be only applied to the patients with HMV who lived with incentive family caregivers in the big city area, because of the limited number of respondents.

6. PLOS authors have the option to publish the peer review history of their article (what does this mean?). If published, this will include your full peer review and any attached files.

Reviewer #1: No

Reviewer #3: No

Reviewer #4: No

---

## [Author Response · Author response to Decision Letter 0]

18 Jun 2021

We thank you and the reviewers for your thoughtful suggestions and insights. The manuscript has benefited from these insightful suggestions. The manuscript has been rechecked and the necessary changes have been made in accordance with the reviewers’ suggestions.

Reviewer #1

Comment 1: This study has used several questionnaires or scales as the measurement tool, for example, the questionnaires for general characteristics of participants including care related variables (line 96-100), Knowledge of Emergency Scale (line 101-102), the ADLs (line 118-119) and the Caring Behavior Scale (line 123-128). However, I cannot find the questionnaires themselves and if the questionnaires have been validated after reading some of the references.

Response 1: We attached the main questionnaires as supporting information. Knowledge of Emergency Scale: supplementary 1 (p.6, line 108) the ADLs: supplementary 2 (p.6, line 125), Caring Behavior Scale: supplementary 3 (p.7, line 137)

Comment 1-2: Although the questionnaires might be published in other journals, the questionnaires might not be suitable for the participants in this study, and the reliability and validity should be mentioned in this article. 

Response 1-2: References which validated the questionnaires are presented accordingly in the manuscript, though mostly only Cronbach’s α was presented in the manuscript. (p.6~7)

Comment 1-3: Although the questionnaires might be published in other journals, the questionnaires might not be suitable for the participants in this study, and the reliability and validity should be mentioned in this article. 

Response 1-3: References which validated the questionnaires are presented accordingly in the manuscript, though mostly only Cronbach’s α was presented in the manuscript. (p.6~7)

Comment 1-4: For example, I am wondering how does the questionnaire ask about the “education for discharge care”. Does the answer “YES” “NO” really mean the true results that the researcher wants to measure?

Response 1-4: As you pointed, “Yes” “No” questions might not be the best option for measuring. However, this questionnaire was selected as it seemed to be the most reliable among the published measures.

Comment 1-5: In addition, the rating method of ADLs seems not the same as the reference [22] mentioned.

Response 1-5: Rating method of ADLs was rechecked and confirmed. (p.6)

Comment 2: How did the 100 participants been selected? Selection issue should be considered.

Response 2: Selection criteria and process were elaborated in the manuscript. (p. 5)

Comment 3: The results in line 216 to line 22 should be re-written without directly mention the beta and p-value which the reader can see in Table 5.

Response 3: As you recommended, we rewrote the results without beta and p-value in the manuscript. (p.14)

Reviewer #3

Question 1: 91.6% of patients did not have experience of emergency situation in the study. However, the table 5 showed the factor of experience of emergency situation positively associated with caring behavior. How could author explain the result because most of patients seem to be relatively stable?

Response 1: Though the percentage of patients who had experienced an emergency situation in the past was small and most of the patients seemed stable as you commented, we analyzed this variable as it had been pointed as an important factor in a previous study. We included this point in Discussion section with reference. (p.18)

Question 2: The mean age of patients is young (38.08±26.62) for the need of HMV. The authors should state what kinds of neuromuscular and airway disease of patients.

Response 2: The mean age of patients is young as the patients included children. Disease of patients were described in the manuscripts. (p.10)

Question 3: In line 194: “Caring behavior of ventilator management was 61.94 (±6.94) out of 50”. It seems 61.94 (±6.94) out of 70 instead of 50. 

Response 3: The values range was corrected both in the manuscript and table 3. (p.12)

Reviewer #4

<Major points>

Comment 1: The conclusion part in the page 17 is too subjective because there was no result in this study support using the simulation training with specific scenarios. I would suggest only the first sentence to be kept. Please make point-to-point reply to the results by Tables.

Response 1: Thanks for the suggestion. We decided to keep the first sentence only as the reviewer suggested. (p.18)

Comment 2: There is one measure needed to be clarified. In Table 3 (Page 11). Caring Behavior Scores of Family Caregivers regarding Ventilator management: 61.94±6.94 (min 14 and Max 50). Why?

Response 2: The values range was corrected both in the manuscript and table 3. (p.12)

Comment 3: For readers, it would be interesting to understand the current caregiver training programs before discharge. Could the authors explain more about this? That would help why the score regarding rehabilitation and communication are relatively lower than other domains. And from the results, what kind of education or training program needs to be modified? Please explore more on this for implication.

Response 3: The current caregiver training programs before discharge were explained in Introduction (p.4). Needs for a specified education were included in Conclusion (p.18).

Comment 4: Could the authors disclose the 12.7 items of required equipment in Table 2?

Response 4: The 12.7 items were added in the manuscript. (p.10)

<Minor points>

Comment 1: This is a self-reported questionnaire to reflect the caring behavior score. I wonder if the actual caring skills of family caregivers can be measured by home visited nurses or health care workers at home.

Response 1: The actual caring skills of family caregivers were not measured by home visited nurses or health care workers at home. We added this point in discussion according to your comment.

Comment 2: This could be only applied to the patients with HMV who lived with incentive family caregivers in the big city area, because of the limited number of respondents.

Response 2: Discussion was modified as the results could be applied for family caregivers in the big city area as the reviewer commented. (p.18)

---

## [Decision Letter · Decision Letter 1]

8 Jul 2021

Factors associated with caring behaviors of family caregivers for patients receiving home mechanical ventilation with tracheostomy: A cross-sectional study

PONE-D-21-02293R1

Dear Dr. YANG,

We’re pleased to inform you that your manuscript has been judged scientifically suitable for publication and will be formally accepted for publication once it meets all outstanding technical requirements.

Kind regards,

Tai-Heng Chen, M.D.

Academic Editor

PLOS ONE

Reviewers' comments:

Reviewer's Responses to Questions

**Comments to the Author**

1. If the authors have adequately addressed your comments raised in a previous round of review and you feel that this manuscript is now acceptable for publication, you may indicate that here to bypass the “Comments to the Author” section, enter your conflict of interest statement in the “Confidential to Editor” section, and submit your "Accept" recommendation.

Reviewer #1: All comments have been addressed

Reviewer #3: All comments have been addressed

Reviewer #4: All comments have been addressed

2. Is the manuscript technically sound, and do the data support the conclusions?

Reviewer #1: Yes

Reviewer #3: Yes

Reviewer #4: Yes

3. Has the statistical analysis been performed appropriately and rigorously? 

Reviewer #1: Yes

Reviewer #3: Yes

Reviewer #4: Yes

4. Have the authors made all data underlying the findings in their manuscript fully available?

Reviewer #1: Yes

Reviewer #3: Yes

Reviewer #4: Yes

5. Is the manuscript presented in an intelligible fashion and written in standard English?

Reviewer #1: Yes

Reviewer #3: Yes

Reviewer #4: Yes

6. Review Comments to the Author

Reviewer #1: Thank you for the opportunity to review this descriptive analysis investigating the factors associated with caring behaviors of family caregivers for patients receiving home mechanical ventilation with tracheostomy. AND congratulation to your work. The issues I have mentioned have been addressed. Please make sure that all supporting information has been uploaded as the supplementary carefully.

Reviewer #3: (No Response)

Reviewer #4: The authors replied my concerns one by one clearly. It looked sound after revision. I have no more question.

7. PLOS authors have the option to publish the peer review history of their article (what does this mean?). If published, this will include your full peer review and any attached files.

Reviewer #1: No

Reviewer #3: No

Reviewer #4: No

---

## [Editor Report · Acceptance letter]

12 Jul 2021

PONE-D-21-02293R1 

Factors associated with caring behaviors of family caregivers for patients receiving home mechanical ventilation with tracheostomy: A cross-sectional study 

Dear Dr. Yang:

I'm pleased to inform you that your manuscript has been deemed suitable for publication in PLOS ONE. Congratulations! Your manuscript is now with our production department. 

Kind regards, 

on behalf of

Dr. Tai-Heng Chen 

Academic Editor

PLOS ONE